# Global Analysis of the Human RNA Degradome Reveals Widespread Decapped and Endonucleolytic Cleaved Transcripts

**DOI:** 10.3390/ijms21186452

**Published:** 2020-09-04

**Authors:** Jung-Im Won, JaeMoon Shin, So Young Park, JeeHee Yoon, Dong-Hoon Jeong

**Affiliations:** 1Smart Computing Lab., Hallym University, Chuncheon 24252, Korea or jiwon@hanyang.ac.kr (J.-I.W.); shin@dbcls.rois.ac.jp (J.S.); 2Center for Innovation in Engineering Education, Hanyang University, Seoul 04763, Korea; 3Database Center for Life Science, Joint Support-Center for Data Science Research, Research Organization of Information and Systems, Kashiwa-Shi, Chiba-Ken 277-0871, Japan; 4Department of Life Science and Multidisciplinary Genome Institute, Hallym University, Chuncheon 24252, Korea; soyoungp@hallym.ac.kr; 5School of Software, Hallym University, Chuncheon 24252, Korea

**Keywords:** mRNA decay, parallel analysis of RNA ends, XRN1, decapping

## Abstract

RNA decay is an important regulatory mechanism for gene expression at the posttranscriptional level. Although the main pathways and major enzymes that facilitate this process are well defined, global analysis of RNA turnover remains under-investigated. Recent advances in the application of next-generation sequencing technology enable its use in order to examine various RNA decay patterns at the genome-wide scale. In this study, we investigated human RNA decay patterns using parallel analysis of RNA end-sequencing (PARE-seq) data from *XRN1*-knockdown HeLa cell lines, followed by a comparison of steady state and degraded mRNA levels from RNA-seq and PARE-seq data, respectively. The results revealed 1103 and 1347 transcripts classified as stable and unstable candidates, respectively. Of the unstable candidates, we found that a subset of the *replication-dependent histone* transcripts was polyadenylated and rapidly degraded. Additionally, we identified 380 endonucleolytically cleaved candidates by analyzing the most abundant PARE sequence on a transcript. Of these, 41.4% of genes were classified as unstable genes, which implied that their endonucleolytic cleavage might affect their mRNA stability. Furthermore, we identified 1877 decapped candidates, including *HSP90B1* and *SWI5,* having the most abundant PARE sequences at the 5′-end positions of the transcripts. These results provide a useful resource for further analysis of RNA decay patterns in human cells.

## 1. Introduction

The gene expression levels within a cell are determined by the interplay of tightly regulated processes for RNA synthesis and decay. Although transcription-mediated mRNA synthesis plays a major role, mRNA decay pathways at the posttranscriptional level also significantly contribute to regulating gene expression. Cytoplasmic mRNAs are internally degraded by endoribonucleases or from one of the extremities by exoribonucleases. These ribonucleolytic activities include decapping, 5′ to 3′ exonucleolytic decay, deadenylation, 3′ to 5′ exonucleolytic decay, and endonucleolytic cleavage [1,2,3,4,5]. The majority of cytoplasmic mRNA decay is initiated by a deadenylation-dependent pathway, which shortens the mRNA 3′ poly(A) tail [2]. This process is mediated by the activity of deadenylase complexes (i.e., CCR4–CAF1–NOT1) or PARN [2,6]. After deadenylation, mRNAs are further degraded by either the 5′ to 3′ or 3′ to 5′ decay pathways [7,8]. A decapping complex typically containing DCP2 hydrolyzes the 5′ cap to expose the mRNA to the 5′ to 3′ exoribonuclease XRN1. Alternatively, the deadenylated mRNA is degraded by the exosome complex in the 3′ to 5′ direction, with the remaining 5′ cap hydrolyzed by the scavenger-decapping enzyme DcpS. For some mRNAs, decay is initiated by endonucleolytic cleavage in a deadenylation-independent mRNA decay pathway that does not involve prior deadenylation [4]. Instead, endonucleolytic cleavage occurs within the body of the mRNA, followed by the degradation of the downstream fragment by XRN1 and the upstream fragment by the exosome. Endoribonucleases, such as AGO, SMG6, and RRP44/DIS3, are involved in this pathway [9]. Nonsense-mediated decay (NMD) is another type of deadenylation-independent mRNA decay pathway [10,11]. NMD targets that typically contain premature termination codons bypass deadenylation and undergo 5′ cap removal by DCP2 and initiated by NMD factors, followed by 5′ to 3′ degradation by XRN1.

RNA decay on a transcriptome-wide scale has been studied while using several methods. One well-established method takes advantage of inhibitor-mediated global transcriptional arrest. In this method, global transcription is blocked by transcription inhibitors, such as actinomycin D, 5,6-dichloro-1–D-ribofuranosyl-benzimidazole and α-amanitin [12], with subsequent RNA decay monitored by gene-expression microarray or RNA-seq [13,14,15]. Additionally, inhibitor-free technologies have been developed in order to determine RNA stability in mammalian cells at the genome-wide scale. In these methods, newly synthesized RNAs are pulse-labeled by uridine analogs, such as 4-thiouridine, 5-etyniluridine, and 5′-bromo-uridine, followed by the purification of labeled de novo RNAs [16,17,18,19]. Streptavidin-mediated separation allows separation of cellular RNA into labeled, newly transcribed RNA and unlabeled, preexisting RNA. Combining this metabolic labeling with genome-wide RNA profiling approaches, such as microarray or RNA-seq, enables genome-wide measurement of RNA transcription and decay in parallel.

Parallel analysis of RNA ends (PARE) or RNA degradome sequencing is another method adopted for RNA decay analysis [20,21,22]. PARE-seq was originally developed in order to identify miRNA-mediated target mRNA decay in plants and has been used to identify widespread miRNA-dependent endonucleolytic cleavage in animals [23,24,25,26]. Because PARE-seq profiles mRNA decay byproducts with mono-phosphorylated 5′ ends and polyadenylated 3′ ends, it is widely used to identify not only miRNA targets, but also targets of mRNA NMD by SMG6 and endoribonucleolytic cleavage of N^6^-methyladenosine-containing RNAs by the RNase P/MRP complex [27,28].

In this study, we analyzed PARE-seq and RNA-seq data that were established from *XRN1*-knockdown HeLa cell lines to investigate global mRNA stability. Using genome-wide bioinformatics analysis, we classified stable and unstable genes and identified endonucleolytically cleaved and decapped candidate transcripts.

## 2. Results

### 2.1. Global Analysis of mRNA Stability in XRN1-Knockdown HeLa Cells

We analyzed a previously published dataset that included PARE-seq and RNA-seq data from *XRN1*-knockdown HeLa cells in order to investigate global RNA stability in human cells [28]. The PARE-seq profiles included decapped and poly(A)-containing mRNA decay products. For the PARE-seq data, the abundance of all PARE sequences matching a specific cDNA was summed and normalized in order to assign a decay reads per kilobase in million (DPKM) reads value representing an mRNA decay level [29]. Additionally, fragments per kilobase in million (FPKM) reads values from RNA-seq data were obtained for steady state mRNA levels (Figure 1A). We compared the DPKM and FPKM values of two biological replicates of PARE-seq and RNA-seq data, respectively, in order to determine data reproducibility. This analysis revealed a high correlation in DPKM and FPKM values between the two biological replicates (RNA-seq R^2^ = 0.999; and PARE-seq R^2^ = 0.704), indicating the high quality and reproducibility of the data (Figure 1B). The stability of the annotated transcripts was then analyzed by comparing the DPKM and FPKM values of each transcript. Using the EdgeR program, we obtained fold changes between DPKM and FPKM values and false-discovery rates (FDRs). Moreover, we assumed that higher DPKM than FPKM values for a transcript indicated its instability, and lower DPKM than FPKM values implied transcript stability. We discovered a positive correlation between DPKM and FPKM values (R^2^ = 0.484) (Figure 1C). We adopted three filters (abundance, significance, and fold change) (Figure 1A) to identify stable and unstable transcripts with high stringency, resulting in the identification of a total of 1103 stable transcripts and 1347 unstable transcripts (Figure 1C and Appendix A).

The set of genes was subjected to Gene Ontology (GO) analysis in order to understand the biological processes associated with these stable and unstable transcripts. We observed that the stable gene set showed a high enrichment for genes that were involved in nucleic acid metabolism, RNA metabolic process, and gene expression (Figure 2A). By contrast, the unstable gene set showed high enrichment for genes related to chromatin silencing, negative regulation of epigenetic gene expression, and chromatin silencing of rDNA (Figure 2B). This result implied that RNA stability is involved in regulating the proper gene expression that is involved in these processes.

We then examined the decay plots (D-plots) and RNA-seq read coverage of the representative stable and unstable transcripts (Figure 3). D-plots represent the mRNA decay patterns by plotting the normalized abundance of PARE sequences matching a transcript versus their location on a cDNA. D-plots of the stable transcripts represented less PARE sequence reads across the transcripts as compared with their RNA-seq read coverages. We then selected the stable genes *LARP1*, *PABPN1*, *BMI1,* and *PTEN*, with their RNA-seq read coverages and PARE-seq D-plots being shown in Figure 3A. *LARP1* and *PABPN1* both encode RNA-binding proteins and they are involved in gene expression by promoting translation or protecting mRNA poly(A) tails from degradation, respectively. *BMI1* is a proto-oncogene encoding a ring protein that is a major component of polycomb repressive complex 1, and *PTEN* encodes a phosphatase that acts as a tumor suppressor. As unstable genes, we selected *CIRBP*, *PDRG1*, *IER3*, and *HIST2H4A* (Figure 3B). *CIRBP* encodes a cold-inducible RNA-binding protein, *PDRG1* plays a role in chaperone-mediated protein folding, *IER3* protects cells from apoptosis (its transcripts are rarely detected), and *HIST2H4A* encodes a histone H4 protein that is responsible for nucleosome structure.

### 2.2. A Subset of the Replication-Dependent Histone Transcripts Is Polyadenylated and Rapidly Degraded

Interestingly, *HIST2H4A*, which encodes a replication-dependent histone (RDH), was identified as an unstable gene (Figure 3B). Given that *RDH* mRNAs have conserved stem-loop structures instead of poly(A) tails, their expression could not be detected in the poly(A)-selected RNA-seq and PARE-seq data. However, we noticed that polyadenylated and degraded *HIST2H4A* mRNA was well represented in PARE-seq data, but barely detected by RNA-seq. Recent reports indicated that a subset of histone genes produces poly(A) mRNAs under a variety of cellular conditions and stress conditions [30,31,32]. Our data indicated that a subset of *RDH* mRNAs was polyadenylated and rapidly degraded in *XRN1*-knockdown HeLa cells. Indeed, many *RDH* transcripts were classified as unstable genes and enriched in the chromatin-silencing GO term.

We examined PARE-seq and RNA-seq data of all histone transcripts, including 60 *RDH* and 13 *replication-independent histone* (*RIH)* transcripts, in order to globally investigate the RNA stability of the histone transcripts. As expected, most *RDH* transcripts were detected at lower levels in poly(A)-selected RNA-seq data, whereas *RIH* transcripts were well represented (Figure 4A). By contrast, some *RDH* and most *RIH* transcripts were detected in PARE-seq data. The relative fold changes between DPKM and RPKM values of histone mRNAs showed that *RDH* transcripts tended to accumulate more decay products, whereas the *RIH* transcripts accumulated fewer decay products (Figure 4B). D-plots of the selected histone transcripts showed that decay products were evenly distributed across the transcripts, although there were distinct peaks at particular positions (Figure 4C–G). The most abundant decay products were identified in either coding regions or untranslated regions, implying that there might be various endoribonucleolytic cleavage pathways in these histone transcripts.

### 2.3. Identification of Endonucleolytically Cleaved Transcripts

Previous studies show that D-plots from PARE-seq data well-represent the decay patterns, such as miRNA-mediated target mRNA cleavage and mRNA NMD, of transcripts [20,28,33]. Among the decay products, endonucleolytically cleaved transcripts tend to be more prominent than other decay byproducts; therefore, we speculated that transcripts with the most abundant PARE-sequence mapping on a cDNA would be by endonucleolytically cleaved transcripts, although we cannot rule out the possibility that hindrance of 5′ to 3′ exonucleolytic decay might also promote high accumulation of decay products.

We identified the most abundant PARE sequence on a transcript, designated as a Max PARE sequence (Max-seq), for all annotated human transcripts to identify endonucleolytically cleaved transcripts. For better identification, we adopted four stringent filters: reproducible Max-seq position, reliable decay abundance, Max-seq abundance, and the more prominent Max-seq value relative to that of other decay byproducts (Figure 5A). By analyzing two biological replicates of PARE-seq data, we identified 380 potential endonucleolytically cleaved transcripts (Appendix A), and their Max-seq was identified at the same positions as the biological replicates. Additionally, the sum of the PARE-sequence abundances (SOAs) on a transcript and the Max-seq abundance were >50 and >10, respectively, and the Max-seq abundance was >20% for all PARE sequences that were mapped on a cDNA. Notably, the relatively high number of transcripts tended to have reproducible Max-seq positions close to the 5′ ends of the cDNAs (Figure 5B), possibly due to the effect of *XRN1* knockdown, which results in defects in 5′ to 3′ exonucleolytic decay. Furthermore, 380 endonucleolytically cleaved-transcript candidates were derived from 184 genes, of which 77 (41.8%) were classified as unstable according to our analysis. We found that no stable genes contained endoribonucleolytic cleavage sites, implying that endonucleolytic cleavage of the 77 genes might affect mRNA stability.

We then selected six transcripts in order to visualize their decay patterns by D-plots (Figure 6). The Max-seqs were identified in various positions of the cDNAs and, notably, the second most abundant PARE sequences were found downstream of the Max-seq position. *Nuclear speckle splicing regulatory protein 1* encodes an RNA-binding protein that mediates pre-mRNA alternative-splicing regulation (Figure 6A). The *CD164* gene contains a potential endoribonucleolytic cleavage site at the 3′ untranslated region (UTR) (Figure 6B) and encodes a transmembrane sialomucin and cell-adhesion molecule that plays a key role in hematopoiesis. Additionally, the Max-seq of *carboxylesterase 2* was located in the 3′ UTR (Figure 6C), with this gene product being involved in xenobiotic detoxification. Multiple alternatively spliced transcript variants of *TNF receptor superfamily member 25* have been reported, five of which have the same Max-seq values identified in the present analysis (Figure 6D). Moreover, *T cell immune regulator 1* encodes multiple isoforms, five of which have the same Max-seq on their coding sequences (Figure 6E), whereas we identified only one *sorting nexin 2* isoform as an endonucleolytically cleaved transcript (Figure 6F).

### 2.4. Identification of Decapped Transcripts

The process of mRNA decapping involves the hydrolysis of the 5′ cap structure and exposure of the 5′ monophosphate by the decapping enzyme DCL2. These decapped mRNAs are further processed by XRN1 and quickly decayed. We assumed that our PARE-seq data could be used to identify decapped transcripts, because the PARE-seq libraries were constructed from *XRN1*-knockdown cell lines and PARE-seq profiles at the 5′ ends of monophosphorylated mRNAs. A prerequisite to identifying decapped transcripts involves the determination of the precise 5′ end positions of mRNAs. To this end, we took advantage of C-PARE-seq, which profiles the 5′ ends of the capped transcript. C-PARE-seq data were established from the same RNAs used in PARE-seq library construction [28].

C-PARE sequences were mapped to 200-nt regions upstream and downstream of the annotated cDNAs (+/− 200-nt windows) in order to identify precise 5′-end positions of mRNAs, resulting in the identification of 41,718 transcripts with at least one C-PARE sequence. Empirical cap positions were defined using three filters (Figure 7A). First, 17,828 transcripts had the most abundant C-PARE sequences (C-PARE Max-seq) at the same position between the two biological replicates. The second filter identified 9269 transcripts that contained reliable C-PARE Max-seqs with >10 reads per million (RPM). The cap position was then identified for 8499 transcripts, in which the ratios of the C-PARE Max-seq abundances and all C-PARE abundances were >0.2. We examined the distribution of C-PARE max sequences across the +/− 200-nt windows in order to compare the 8499 validated 5′ ends and the annotated 5′ ends of the hg19 human reference genome (Figure 7B). We found that 10.5% (890/8499) of C-PARE Max-seqs exactly matched the annotated 5′ ends of the transcripts. Moreover, 73.6% (6254/8499) of transcripts had their C-PARE Max-seqs within 50-nt upstream and downstream of the annotated cDNAs, implying that the C-PARE Max-seqs represented the 5′-cap positions of the cDNAs.

PARE sequences were mapped to +/− 200-nt windows to identify decapped transcripts, obtaining 85,178 transcripts with at least one PARE sequence. These were further filtered using the same criteria used for C-PARE analysis. The results identified Max-seqs within these windows for 3852 transcripts. We then compared the positions of C-PARE Max-seq and PARE Max-seq in each transcript, identifying 1877 transcripts as decapped candidates based on their C-PARE Max-seqs and PARE Max-seqs being at the same positions (Appendix A). These 1877 decapped candidate transcripts corresponded to 572 genes, of which 207 (36.2%) and eight (1.4%) were classified as unstable and stable genes, respectively. These results suggested that the decapping process of these 207 genes might significantly affect their mRNA stability.

We then performed GO analysis of the 1877 decapped candidate transcripts. The set was highly enriched for genes that were involved in the mRNA metabolic process, nuclear-transcribed mRNA catabolic process, viral gene expression, and NMD (Figure 8A). In the cellular component, genes in the intracellular organelle, intracellular organelle part, and cytoplasmic part were enriched (Figure 8B), and those with molecular functions that were associated with protein binding, nucleic acid binding, and RNA binding were enriched (Figure 8C).

The generation of D-plots for the decapped candidates indicated that within the +/− 200-nt window, the position of the 5′ cap was predicted near the annotated 5′ end of cDNA. Of the six transcripts that are shown in Figure 9, two showed the exact same positions of the 5′ end of cDNAs (Figure 9E,F). In general, the abundance of C-PARE Max-seq was much higher than that of the other C-PARE abundances. Additionally, the PARE Max-seq, which represents the decapped transcript, was followed by several abundant PARE sequences that are supposedly decay byproducts following decapping. *HSP90B1-001* and *RPL11-001* are examples of transcripts categorized as stable genes (Figure 9A,B). The others (*HDAC5-008*, *SWI5-001*, *LDHA-018*, and *FTH1-005*) were categorized as unstable genes (Figure 9C–F) and also identified among endoribonucleolytically cleaved candidates. These findings indicated that the most abundant PARE sequences within the +/− 200-nt window were also the most abundant PARE sequences mapped on these cDNAs.

## 3. Discussion

RNA decay mechanisms play an important role in regulating gene expression. Although the molecular function of RNA decay machinery has been well-characterized, the global identification of its regulated target RNAs has been relatively less reported. Here, we identified potential mRNA decay targets in humans by analyzing mRNA decay byproducts. By comparing PARE-seq and RNA-seq data, we classified genes as stable or unstable and identified candidates of endonucleolytically cleaved transcripts and decapped transcripts by analyzing the most abundant PARE sequence on each transcript.

Sequence abundance according to RNA-seq and PARE-seq data represents the steady state and degradation levels of each mRNA, respectively. Previous reports show that the extent of RNA stability in plants can be monitored by comparing the FPKM value of RNA-seq data and DPKM value of PARE-seq data [29,34,35]. This kind of application to the best of our knowledge is less reported in animals, except for the identification of the targets of specific mRNA decay machinery, such as the RNase P/MRP complex and SMG6 [27,28]. Previous genome-wide analyses of animal mRNA stability have been mostly conducted by monitoring gene expression while using microarray or RNA-seq data after treatment with transcription inhibitors or metabolic labeling of nascent RNAs [13,14,15,16,19]. In the present study, comparison of PARE-seq and RNA-seq data from *XRN1*-knockdown HeLa cells allowed for the identification of specific categories of gene sets enriched in stable and unstable gene candidates. Genes involved in nucleic acid metabolism, RNA metabolism, and gene expression were highly enriched in the stable gene set, whereas genes that were related to chromatin silencing, negative regulation of epigenetic gene expression, and chromatin silencing of rDNA were highly enriched in the unstable gene set (Figure 2). Our analysis revealed that *PTEN* transcripts had higher RNA-seq reads than PARE-seq reads, suggesting *PTEN* as a stable gene (Figure 3A). Previous studies identified *PTEN* as among the most frequently mutated tumor-suppressor gene in human cancer, with its levels being negatively regulated by miRNAs, such as miR-21 and miR-130 family members [36,37]. This implies that negative regulation of *PTEN* expression is compromised in *XRN1*-knockdown HeLa cells. Conversely, *CIRBP* encoding a cold-inducible RNA-binding protein was identified as an unstable gene, because its transcripts were rarely detected in RNA-seq data, whereas its degraded transcripts were highly detected in PARE-seq data (Figure 3B). Because CIRBP modulates circadian gene expression posttranscriptionally, further analysis is required in order to elucidate how *CIRBP* is posttranscriptionally regulated [38].

Of the identified unstable genes, many *RDHs* accumulated polyadenylated and degraded transcripts, because RNA-seq cannot detect non-poly(A)-tailed canonical *RDH* transcripts. *RDH* mRNAs are non-polyadenylated in metazoans and instead contain their characteristic 3′ UTR with highly conserved stem-loop structures [39]. If *RDH* mRNAs are inappropriately processed at the 3′ ends and contain polyadenylated tails, they are rapidly degraded by AU-rich element-mediated mRNA decay via BRF1 [40]. Under cellular stresses, such as UV-C irradiation or puromycin treatment, BRF1 is degraded, whereas the association of HuR with the 3′ UTR of polyadenlyated *RDH* mRNAs increases [31], thereby stabilizing polyadenylated *RDH* mRNAs. In the present study, we found that most polyadenylated *RDH* mRNAs were unstable along with high accumulation of their decay byproducts. Although we were unable to identify the conserved decay pattern of polyadenylated RDHs, we did identify potential endoribonucleolytic cleavage patterns in various positions of RDH cDNAs (Figure 4). The decay patterns of *RDH* mRNAs presented here will be useful for understanding how polyadenylated *RDH* mRNAs are regulated.

By analyzing the most abundant PARE sequences in each transcript, we identified 380 endonucleolytically cleaved transcripts that accumulate specific decay products relative to other randomly decayed byproducts. Although endonucleolytic cleavage by a specific RNase might be the major event in these transcripts, we cannot rule out other possibilities. For example, specific decay products can accumulate with the hindrance of 5′ to 3′ exoribonucleolytic activity following the binding of RNA-binding proteins or the formation of a structural barrier in the form of an RNA sequence. Because our PARE data were from *XRN1*-knockdown cells, comparative analysis with other PARE data derived from knockout of various endonucleases and RNA-binding proteins could provide detailed decay patterns of endonucleolytically cleaved transcript candidates.

Because decapped transcripts are further degraded by the 5′ to 3′ exoribonucleolytic activity of XRN1, it was unsurprising that we identified a large number of decapped transcripts that result from *XRN1* knockdown. We assumed that our decapped candidates were stabilized decay byproducts after decapping and subsequent to the decreased 5′ to 3′ exoribonucleolytic degradation following *XRN1* knockdown. Of the decapped candidates that are shown in Figure 9, *HSP90B1* mRNAs are reportedly associated with hDCP2 [41], with mRNAs of several heat-shock proteins (HSPs), including HSP70 and HSP90B1, bound by hDCP2. Additionally, *Drosophila HSP70* mRNAs are reportedly regulated by the mRNA decay pathway, which includes deadenylation, followed by cap hydrolysis and 5′ exonucleolytic degradation [42]. Moreover, in *DCP2*-knockout plants, several HSPs were significantly upregulated, indicating a conserved function involving regulation of the HSP gene transcript stability by a decapping-dependent mRNA-decay pathway [43]. Another decapped candidate, *SWI5*, is a mitosis-dependent mRNA-stability switch in yeast, where the mitotic exit network protein Dbf2p binds to *SWI5* mRNAs co-transcriptionally to regulate their decay in the nucleus [44]. Upon export to the cytoplasm, *SWI5* mRNA stability is also maintained by Dbf20p. Furthermore, a signal that is related to cell cycle progression can initiate the decay of *SWI5* mRNAs by CCR4–NOT1-complex-dependent deadenylation [44]. In the present study, our findings implied that *SWI5* mRNA in HeLa cells might be further degraded by a decapping pathway.

## 4. Materials and Methods

### 4.1. PARE-Seq and C-PARE-Seq Data Processing

Previously published PARE and C-PARE-seq data from *XRN1*-knockdown HeLa cells [28] were downloaded from the National Center for Biotechnology Information Gene Expression Omnibus (NCBI GEO) under accession number GSE61398 (GSM1503892, GSM1503898, GSM1503895, and GSM1503901). Data processing was performed, as previously reported [45]. Briefly, two biological replicates of PARE and C-PARE reads were trimmed by removing the 3′-adapter sequences while using the cutadapt program [46]. Of the trimmed reads, those <19 nt were discarded, and reads >20 nt were adjusted to 20 nt by removing the sequences after 20 nt. As a result, only 20-nt reads were obtained. Next, reads with simple sequence repeats (SSRs), reads matching >20 hits to the human reference genome hg19, and reads matching a human mitochondrial genome were discarded while using the Bowtie2 program [47]. SSRs were considered to be sequences containing a continuous 12-nt single, double, or triple repeat. The human reference genome hg19 was provided as a pre-built index in the Bowtie2 program. The human mitochondrial genome was downloaded from MITOMAP [48]. One and zero mismatches were allowed when the PARE sequences were mapped to the human genome and mitochondrial genome, respectively. The final reads were normalized to RPM, and normalized PARE sequences were mapped to human cDNAs while allowing one nucleotide mismatch.

### 4.2. RNA-Seq Data Analysis

RNA-seq data from *XRN1*-knockdown HeLa cells [1] were downloaded from NCBI GEO under accession number GSE61398 (GSM1503904 and GSM1503907). RNA-seq reads were mapped to the human reference genome hg19 using TopHat [49] while allowing one nucleotide mismatch for the 49-nt reads. Cufflinks was used in order to determine transcript abundance [49], and FPKM values were obtained after normalization. 

### 4.3. Comparative Analysis of PARE-Seq and RNA-Seq Data

DPKM was calculated by summing the abundance of all PARE sequences matching a cDNA and normalizing the summed abundance by cDNA length and total read number. The EdgeR program [50] was used to compare the FPKM and DPKM of each transcript. FDR and fold changes (log_2_[FPKM/DPKM]) were adjusted to <0.01 and >|2.5|, respectively.

### 4.4. Computational Analysis of Endonucleolytically Cleaved and Decapped Transcripts

Endonucleolytic cleavage sites and decapped transcripts were identified while using custom Python scripts. The SOA and the abundance of PARE Max-seqs were first obtained after mapping the PARE sequences to the annotated transcript in order to identify endonucleolytic cleavage sites. Annotated transcripts were downloaded from the Gencode V11 (https://www.gencodegenes.org/) protein-coding transcripts. Four filters were adopted (Figure 5). SOA and Max-seq abundances were obtained after mapping the PARE sequences to the upstream and downstream 200-nt regions of the transcription-start sites in order to identify decapped transcripts. Upstream 200-nt sequences were extracted from the genome and concatenated with the first 200-nt sequences of the protein-coding transcripts obtained from Gencode V11. After processing the three filters that are shown in Figure 7, decapped candidates were identified by comparing the position of the C-PARE Max-seq with the PARE Max-seq.

### 4.5. GO Annotation

GO analysis of a set of genes was performed using the DAVID bioinformatics analysis platform [51,52]. A functional annotation chart was generated while using default settings.

## 5. Conclusions

In this study, we analyzed RNA decay byproducts from PARE-seq data and identified the diverse RNA-decay patterns of human genes, including endonucleolytically cleaved and decapped candidates. The findings revealed various RNA decay patterns not only with quantitative information, but also the exact positions of the cleaved sites at single-nucleotide resolution. The list of candidate genes identified in this study will be useful resources for further analysis of their expression. Although the data were limited to those from *XRN1*-knockdown HeLa cells, additional data from wild-type and mutated cells defective in other RNA decay machinery will provide a better understanding of the roles of RNA decay mechanisms in regulating gene expression. Additionally, our analysis can be applied in order to understand the role of mRNA stability in various conditions or among the different tissues.

## Figures and Tables

**Figure 1 ijms-21-06452-f001:**
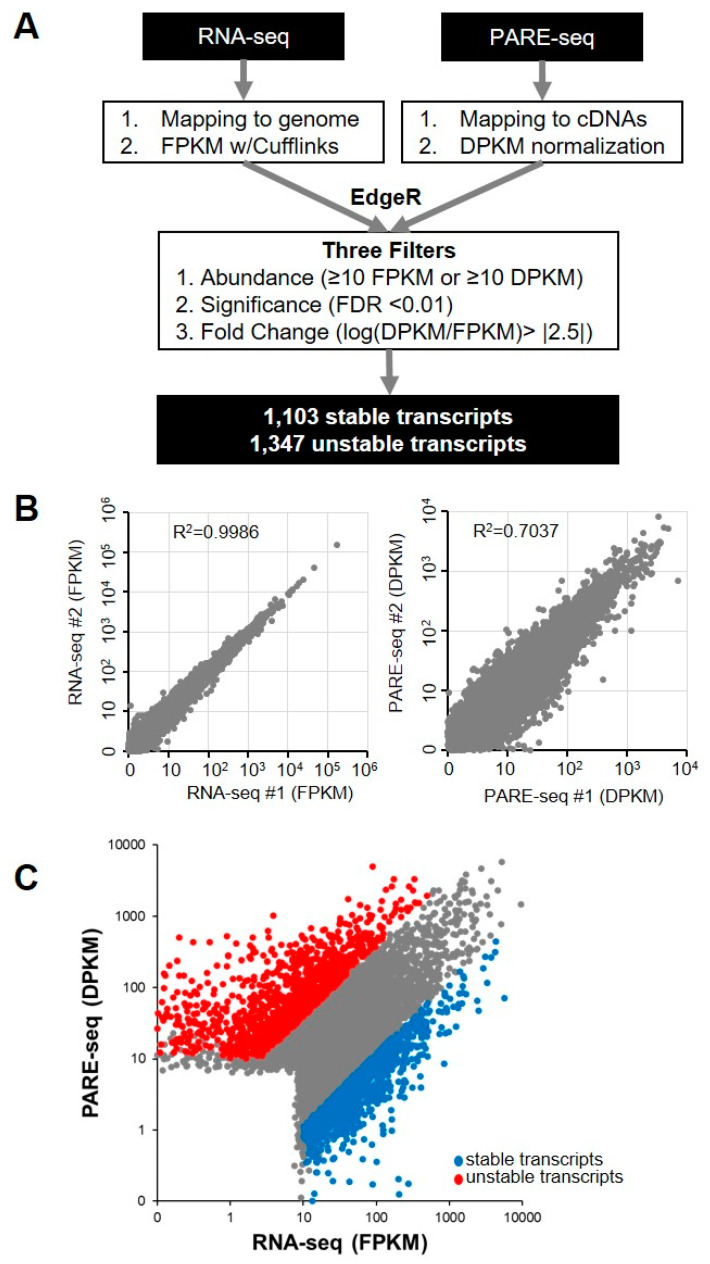
Identification of stable and unstable transcripts. (**A**) Pipeline for the identification of stable and unstable transcripts from RNA-seq and Parallel analysis of RNA ends (PARE)-seq libraries. All sequences from RNA-seq and PARE-seq data were mapped to annotated cDNA and normalized. Stable and unstable transcripts were identified by EdgeR analysis and three filters; (**B**) scatter plots showing the reproducibility of the RNA-seq and PARE seq libraries. Each dot indicates transcripts with abundance values representative of two biological replicate libraries; and, (**C**) scatter plot of transcripts with RNA-seq abundance and PARE-seq abundance values. Blue and red dots indicate stable and unstable transcripts, respectively.

**Figure 2 ijms-21-06452-f002:**
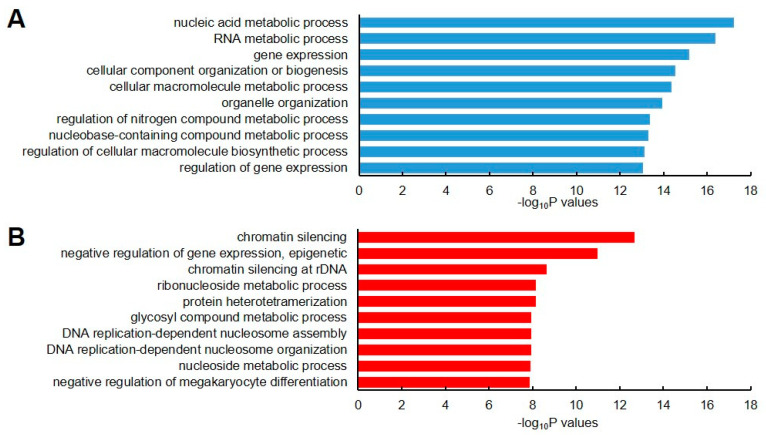
Gene ontology (GO) enrichment analysis of stable and unstable transcripts. The top 10 Biological Process GO terms for stable (**A**) and unstable (**B**) transcripts are represented as blue and red colors, respectively. The top 10 GO terms were selected based on their lower *p*-values.

**Figure 3 ijms-21-06452-f003:**
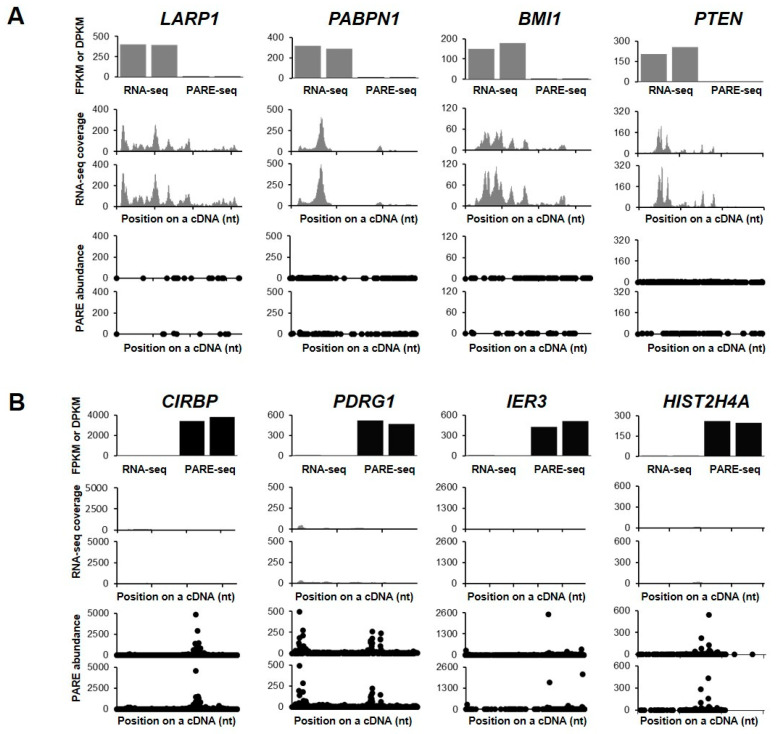
Selected examples of stable and unstable genes. RNA-seq and PARE-seq data for the selected stable (**A**) and unstable (**B**) genes. The first histograms represent the fragments per kilobase in million (FPKM) and decay reads per kilobase in million (DPKM) values of RNA-seq and PARE-seq data, respectively. The second histograms show the coverage of RNA-seq reads on a cDNA. The third plot is a D-plot indicating the decay patterns of PARE-seq data. The graphs are representative of two biological replicates. The X- and Y-axis represent positions on the cDNA and the normalized read depth, respectively. Grey and black colors indicate data from RNA-seq and PARE-seq, respectively.

**Figure 4 ijms-21-06452-f004:**
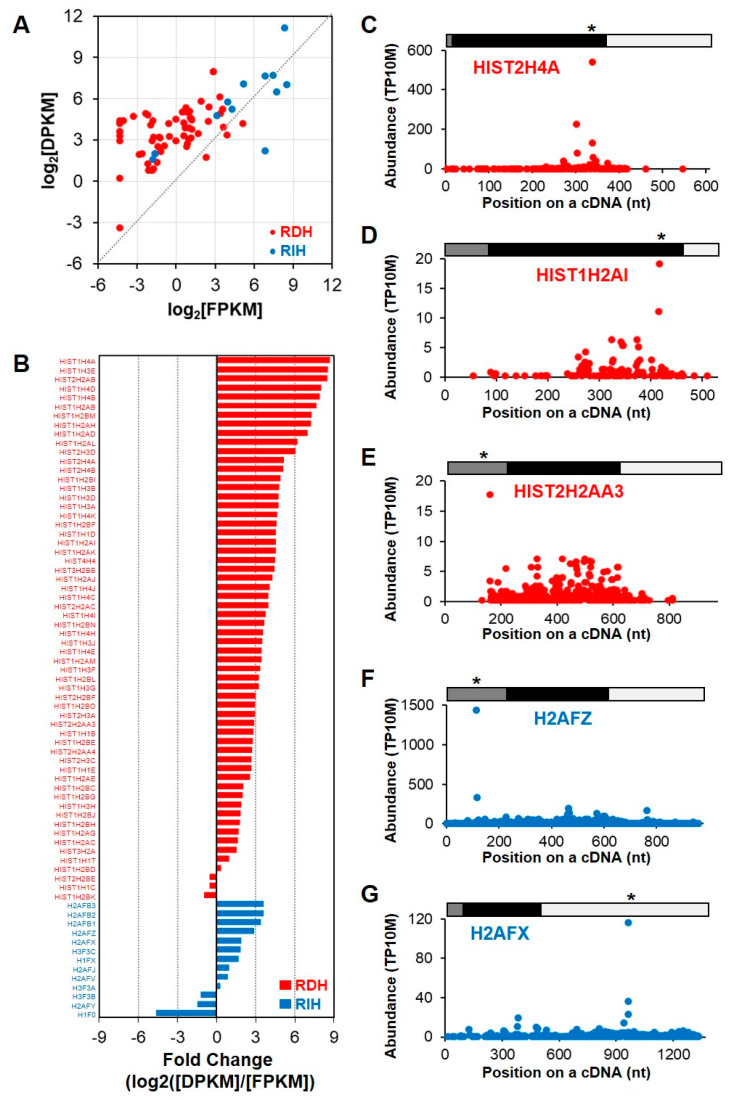
RNA stability of replication-dependent histone (RDH) mRNAs. (**A**) Scatter plot of relative expression levels according to RNA-seq (log2[FPKM]; X-axis) and degraded levels according to PARE-seq (log2[DPKM]; Y-axis). Red and blue dots indicate RDH and replication-independent histone (RIH) mRNAs, respectively; (**B**) relative fold change of RDH and RIH mRNA between expression levels according to RNA-seq data and degraded levels according to PARE-seq data; (**C–G**) and, selected D-plots of RDH and RIH mRNA. The normalized abundances of PARE sequences were plotted on the annotated cDNA positions. Grey, black, and white bars indicate 5′ untranslated region (UTR), coding region, and 3′ UTR, respectively. Star indicates the position of the most abundance PARE sequence. * *p* < 0.05.

**Figure 5 ijms-21-06452-f005:**
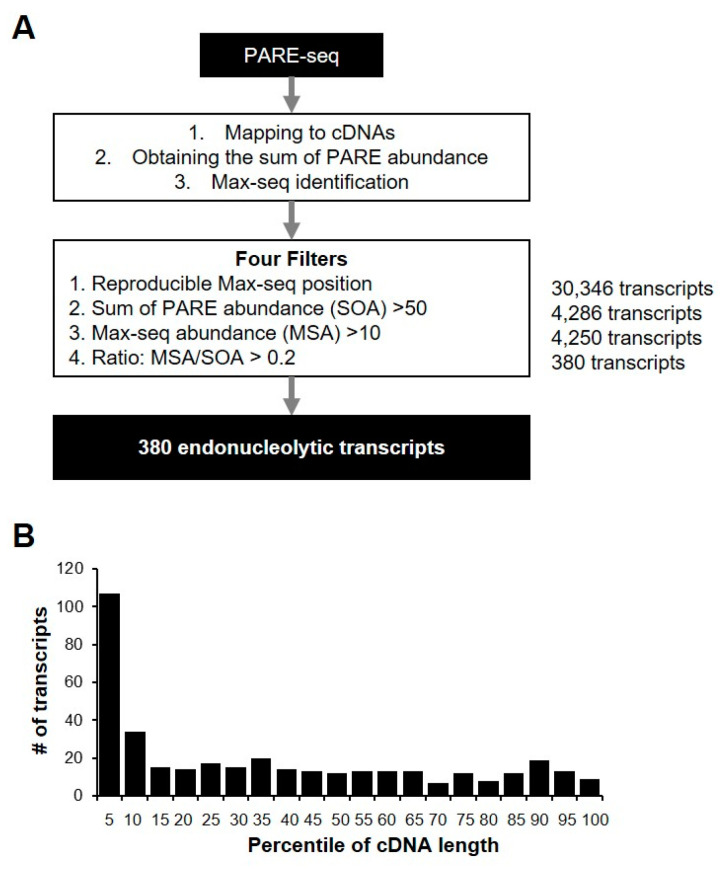
Identification of endonucleolytically cleaved transcripts. (**A**) Pipeline for the identification of endonucleolytically cleaved transcripts from PARE-seq libraries. All of the sequences from PARE-seq data were mapped to annotated cDNA and normalized. Endonucleolytically cleaved transcripts were identified using four filters. The number of transcripts passing each filter is indicated in bold (right); (**B**) relative coverage of each percentile of gene length for the 380 endonucleolytic cleavage sites. The number of transcripts was counted according to the endonucleolytic cleavage position for each percentile of cDNA length.

**Figure 6 ijms-21-06452-f006:**
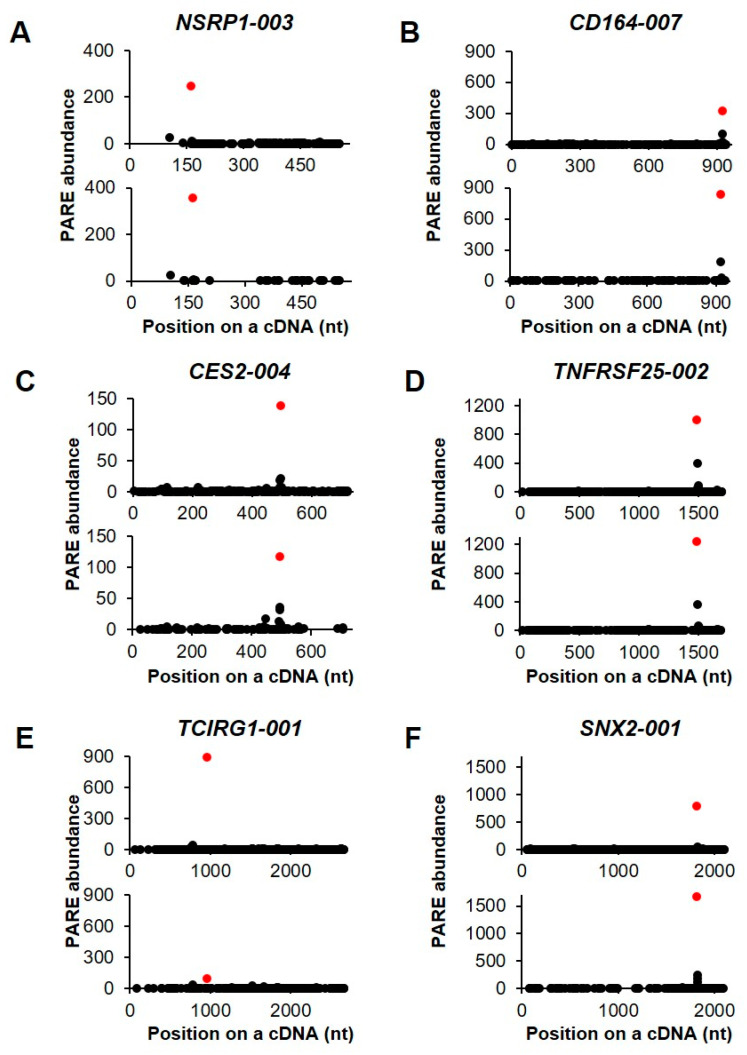
Selected examples of decay plots. (**A**–**F**). D-plots for six endonucleolytically cleaved transcripts and representative of two biological replicates. The normalized abundance of the PARE sequence is plotted on the cDNA position. Red dots indicate the most abundant PARE sequences.

**Figure 7 ijms-21-06452-f007:**
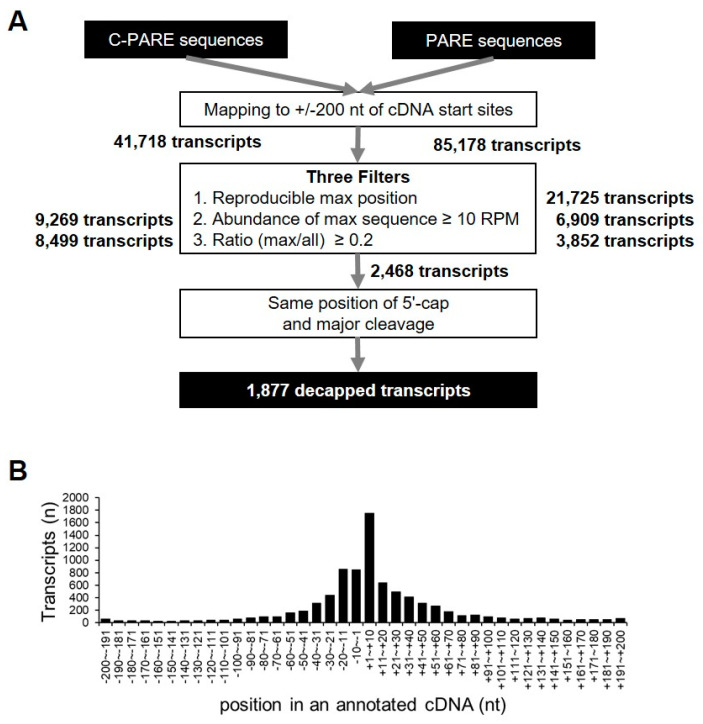
Identification of decapped transcripts. (**A**) Pipeline for the identification of decapped transcripts from C-PARE and PARE libraries. All C-PARE and PARE sequences were mapped to +/− 200-nt regions of annotated cDNA transcription-start sites. The position of the 5′ cap and the major cleavage site in a +/− 200-nt window were identified using three filters and C-PARE and PARE sequences, respectively. The number of transcripts passing each filter is indicated in bold (right). The decapped transcripts were identified when the 5′-cap position and the major cleavage site were the same; (**B**) Distribution of the most abundant C-PARE sequences at a given position within the +/− 200-nt window. Histograms represent the number of genes in which a 5′ cap was identified. The relative position from the annotated transcription-start site of a gene is indicated on the X-axis.

**Figure 8 ijms-21-06452-f008:**
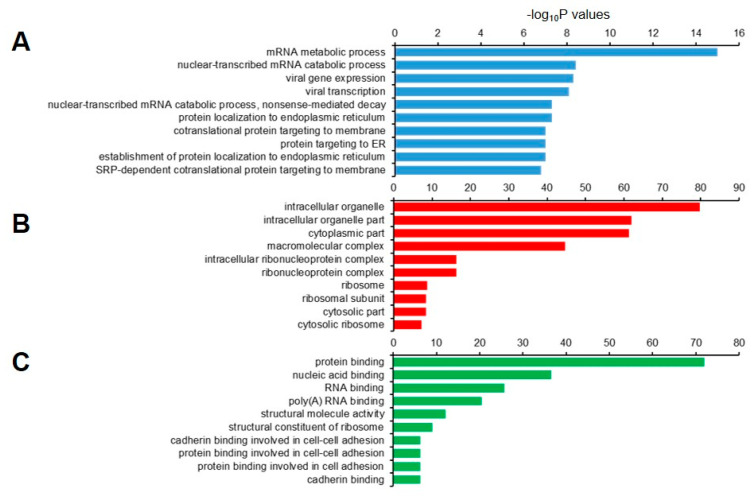
Gene ontology (GO) enrichment analysis of 1,877 decapped transcripts. The top 10 (**A**) biological process; (**B**) cellular component, and (**C**) molecular function GO terns are represented.

**Figure 9 ijms-21-06452-f009:**
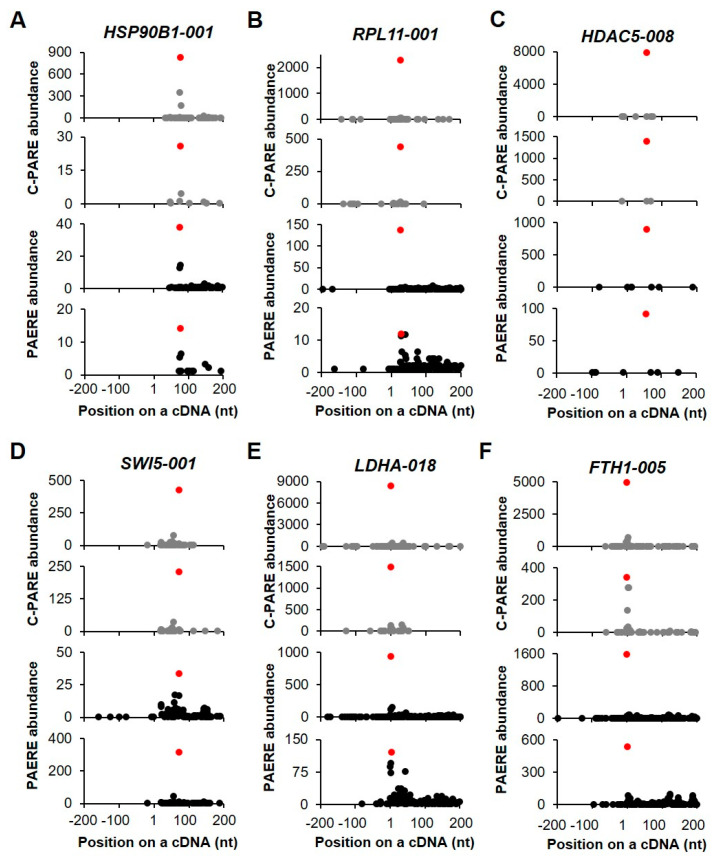
Selected examples of decapped transcripts. (**A**–**F**) The plots show the C-PARE sequence abundance and PARE sequence abundance along the cDNA positions in a +/− 200-nt window from two biological replicates. Red dots indicate the most abundant C-PARE and PARE sequences at the same positions.

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
