# Peer review of "Global Analysis of the Human RNA Degradome Reveals Widespread Decapped and Endonucleolytic Cleaved Transcripts"

_ijms, 2020, doi:10.3390/ijms21186452_

Round 1

Reviewer 1 Report

In this manuscript by Won et al., authors analysed the global RNA decay patterns using the published RNA degradome sequencing and RNA seq database from XRN1 knockdown Hela cells. Using the genome-wide bioinformatic analysis authors classified the results with stable and unstable transcripts and also reported endonucleolytically cleaved, decapped candidates.  They showed that the genes involved in nucleic acid metabolism, RNA metabolism, and gene expression were highly enriched in the stable gene set, whereas genes related to chromatin silencing, negative regulation of epigenetic gene expression, and chromatin silencing of rDNA were highly enriched in the unstable gene sets.

Overall, this manuscript was presented well and the results support the authors claims. This paper provides a good resource to the RNA decay pattern analysis and individual transcript expression.

Major issues:

Authors’ analysis was limited only to one pathway(XRN1 depletion) of RNA degradation, similar genome data analysis of other RNA degradation pathways would show more insights on RNA decay pattern. Ofcourse this point was well acknowledged by authors in their discussion.

Minor issues:

Authors should check and proofread for minor typos and grammar errors.

For eg., Line 176 mapping on a cDNA would by (be) endonucleolytically cleaved transcripts.

Author Response

Response to Reviewer 1 Comments

In this manuscript by Won et al., authors analysed the global RNA decay patterns using the published RNA degradome sequencing and RNA seq database from XRN1 knockdown Hela cells. Using the genome-wide bioinformatic analysis authors classified the results with stable and unstable transcripts and also reported endonucleolytically cleaved, decapped candidates.  They showed that the genes involved in nucleic acid metabolism, RNA metabolism, and gene expression were highly enriched in the stable gene set, whereas genes related to chromatin silencing, negative regulation of epigenetic gene expression, and chromatin silencing of rDNA were highly enriched in the unstable gene sets.

Overall, this manuscript was presented well and the results support the authors claims. This paper provides a good resource to the RNA decay pattern analysis and individual transcript expression.

Major issues:

Authors’ analysis was limited only to one pathway(XRN1 depletion) of RNA degradation, similar genome data analysis of other RNA degradation pathways would show more insights on RNA decay pattern. Ofcourse this point was well acknowledged by authors in their discussion.

Response 1: We appreciate your precious comments on the limitation of our data analysis and results. As you pointed out, we only analyzed the data from XRN1-kwnockdown HeLa cell lines. We believe that future studies using PARE data from the mutants of other RNA degradation pathway, such as decapping, endonucleolytic cleavage, and deadenylation, will provide more resources for RNA decay patterns in humans. We have indicated this limitation and future direction on the lines 335-338 and the lines 405-409. We hope our manuscript can provide useful resource and methods for studying RNA decay patterns.

Minor issues:

Authors should check and proofread for minor typos and grammar errors.

For eg., Line 176 mapping on a cDNA would by (be) endonucleolytically cleaved transcripts.

Response 2: Thank you for this comment. We have looked over our manuscript and corrected typos, including the line 176. In addition, we have attached the certificate of English editing by professional editors at Editage.

Reviewer 2 Report

In the paper entitled “Global Analysis of the Human RNA Degradome  Reveals Widespread Decapped and Endonucleolytic Cleaved Transcripts” Jung-Im Won and colleagues aim to invstigate at the global level human RNA decay patterns taking advantages by the application of next-generation sequencing technology. I have some general and more specific concerns about this work.

My first basic doubt concerns the biological meaning of this approach.

mRNA levels in cells are determined by the relative rates of RNA production and degradation.  The regulation of both mRNA stability and transcription efficiency of different classes of genes is modulated in space, time and as a function of specific stimuli or stress. Therefore, which is the biological sense of analyzing stable and unstable genes in HeLa cells? Indeed, in other genome-wide analyses of animal mRNA stability which are mentioned by the same authors, the role of mRNA stability is investigated to determine the kinetics of response to a specific effect such as DNA damage in a human T cell line (Barenco et al., 2009) or induction of inflammation (Hao and Baltimore, 2009).

Moreover the authors exploit parallel analysis of RNA end-sequencing (PARE-seq) data from XRN1- knockdown HeLa cell lines to investigate human RNA decay patterns. As the authors clearly explain in the nice introduction of the manuscript, XRN1 is a 5' to 3' exoribonuclease involved in several of the mRNA degradation mechanisms. Therefore, the hindrance of this mechanism might affect several of the results obtained by the global analysis as highlighted in several points of the manuscript by the authors themselves.  

If, in my opinion, it is of scarce relevance the presentation of some specific genes identified in a specific class of stable or unstable genes, it might be more interesting to understand the biological processes associated with these stable and unstable transcripts by Gene Ontology (GO) analysis. In chapter 2.1 the authors observed that the stable gene set  showed a high enrichment for genes involved in nucleic acid metabolism and RNA metabolic process while the unstable gene set showed high enrichment for genes related to chromatin silencing. However, in chapter 2.3, after an approach to identify decapped transcripts which lead the authors to conclude that “decapping process of these genes might significantly affect their mRNA stability”, they observed that this set of decapped genes was highly  enriched for genes involved in the mRNA metabolic process, apparently contradicting the results of chapter 2.1.

Author Response

Response to Reviewer 2 Comments

In the paper entitled “Global Analysis of the Human RNA Degradome  Reveals Widespread Decapped and Endonucleolytic Cleaved Transcripts” Jung-Im Won and colleagues aim to invstigate at the global level human RNA decay patterns taking advantages by the application of next-generation sequencing technology. I have some general and more specific concerns about this work.

My first basic doubt concerns the biological meaning of this approach.

mRNA levels in cells are determined by the relative rates of RNA production and degradation.  The regulation of both mRNA stability and transcription efficiency of different classes of genes is modulated in space, time and as a function of specific stimuli or stress. Therefore, which is the biological sense of analyzing stable and unstable genes in HeLa cells? Indeed, in other genome-wide analyses of animal mRNA stability which are mentioned by the same authors, the role of mRNA stability is investigated to determine the kinetics of response to a specific effect such as DNA damage in a human T cell line (Barenco et al., 2009) or induction of inflammation (Hao and Baltimore, 2009).

Response 1: Thank you for pointing out the limitation of our data analysis and our results. As you pointed out, we only analyzed the available data from XRN1-knockdown HeLa cell lines. We were trying to identify the transcripts with stable/unstable status in XRN1-knockdown HeLa cell lines and to understand how these unstable transcripts are regulated by endonucleolytic cleavage and decapping. Given that the expression levels from RNA-seq data represent both intact mRNAs and degraded RNAs whereas the decay levels from PARE-seq data are only from the degraded RNAs, we have adopted a more stringent cutoff (log(DPKM/FPKM) > |2.5|) to classify the stable and unstable transcripts. This kind of application was reported in several papers and we discussed on lines 295 – 299. We agree with your concern that the role of mRNA stability should be investigated to determine the kinetics of responses to various conditions. Therefore, we agreed the limitation of our results and discussed this issue on lines 335 – 338 and lines 405 – 409.   

Moreover the authors exploit parallel analysis of RNA end-sequencing (PARE-seq) data from XRN1- knockdown HeLa cell lines to investigate human RNA decay patterns. As the authors clearly explain in the nice introduction of the manuscript, XRN1 is a 5' to 3' exoribonuclease involved in several of the mRNA degradation mechanisms. Therefore, the hindrance of this mechanism might affect several of the results obtained by the global analysis as highlighted in several points of the manuscript by the authors themselves.

Response 2: Thank you again for your comments on the limitation of our results. We tried our best to investigate RNA stability in XRN1-knockdown HeLa cell lines using the currently available data. As you noticed, the XRN1-knockdown HeLa cells accumulate more RNA decay byproducts than control HeLa cells. Because data from control HeLa cells are not available, we assumed that comparison between steady-state expression levels of mRNAs and accumulated RNA decay in the XRN1-knockdown HeLa cells can provide the information of mRNA stability status. Indeed, as shown in Figure 3, the values of FPKM and DPKM were dramatically different in the selected genes of stable and unstable candidates. Although our stable/unstable candidates may not reflect the authentic mRNA stability of them, some of them, such as CIRBP, were reported as unstable genes in other studies. Thus, we suggested that the list of stable/unstable, endonucleolytic cleaved, decapped candidates, provided as supplemental table 1,2, and 3, will be a useful resource for future analysis of RNA stability in human cells.

If, in my opinion, it is of scarce relevance the presentation of some specific genes identified in a specific class of stable or unstable genes, it might be more interesting to understand the biological processes associated with these stable and unstable transcripts by Gene Ontology (GO) analysis. In chapter 2.1 the authors observed that the stable gene set  showed a high enrichment for genes involved in nucleic acid metabolism and RNA metabolic process while the unstable gene set showed high enrichment for genes related to chromatin silencing. However, in chapter 2.3, after an approach to identify decapped transcripts which lead the authors to conclude that “decapping process of these genes might significantly affect their mRNA stability”, they observed that this set of decapped genes was highly  enriched for genes involved in the mRNA metabolic process, apparently contradicting the results of chapter 2.1.

Response 3: We appreciate your comments on the GO analysis. You pointed out the contradicting results of enriched GO terms between stable genes and decapped genes. Indeed, highly enriched GO terms of stable genes include nucleic acid metabolic process (4719 genes) and “RNA” metabolic process (4249 genes). Highly enriched GO terms of decapped genes include the “mRNA” metabolic process (656 genes) and nuclear transcribed mRNA catabolic process (219 genes). According to the GO tree, nuclear transcribed mRNA catabolic process is a part of mRNA metabolic process. And, mRNA metabolic process is a part of the RNA metabolic process. The GO term, RNA metabolic process, includes not only the mRNA metabolic process but also histone mRNA metabolic process, mRNA cleavage, mRNA modification, mRNA transcription, and so on. This implies that although the genes related to “RNA” metabolic process are enriched in stable genes, specific genes involved in the “mRNA” metabolic process, which is a part of the “RNA” metabolic process, are highly enriched in decapped genes. Indeed, 36.2% of decapped genes were classified as unstable genes, whereas only 1.4% of those were classified as stable genes. Therefore, we concluded that the decapping process of 207 genes might significantly affect their mRNA stability on lines 259-260. We hope these explanations can convince you to believe our results.

Round 2

Reviewer 2 Report

The authors have done their best to compensate for the limitations of the paper that, in my opinion, are important. However I have to admit that there is a lot of work technically well done in this manuscript.